# Comprehensive Characterization of a Novel Bacteriophage, vB_VhaS_MAG7 against a Fish Pathogenic Strain of *Vibrio harveyi* and Its In Vivo Efficacy in Phage Therapy Trials

**DOI:** 10.3390/ijms24098200

**Published:** 2023-05-03

**Authors:** Stavros Droubogiannis, Lydia Pavlidi, Dimitrios Skliros, Emmanouil Flemetakis, Pantelis Katharios

**Affiliations:** 1Institute of Marine Biology, Biotechnology & Aquaculture, Hellenic Centre for Marine Research, 71500 Heraklion, Greece; 2Laboratory of Molecular Biology, Department of Biotechnology, School of Food, Biotechnology and Development, Agricultural University of Athens, 11855 Athens, Greece

**Keywords:** *Vibrio harveyi*, aquaculture, phage therapy, gilthead seabream

## Abstract

*Vibrio harveyi*, a significant opportunistic marine pathogen, has been a challenge to the aquaculture industry, leading to severe economical and production losses. Phage therapy has been an auspicious approach in controlling such bacterial infections in the era of antimicrobial resistance. In this study, we isolated and fully characterized a novel strain-specific phage, vB_VhaS_MAG7, which infects *V. harveyi* MM46, and tested its efficacy as a therapeutic agent in challenged gilthead seabream larvae. vB_VhaS_MAG7 is a tailed bacteriophage with a double-stranded DNA of 49,315 bp. No genes linked with virulence or antibiotic resistance were harbored in the genome. The phage had a remarkably large burst size of 1393 PFU cell^−1^ and showed strong lytic ability in in vitro assays. When applied in phage therapy trials in challenged gilthead seabream larvae, vB_VhaS_MAG7 was capable of improving the survival of the larvae up to 20%. Due to its distinct features and safety, vB_VhaS_MAG7 is considered a suitable candidate for applied phage therapy.

## 1. Introduction

*Vibrio harveyi* is one of the most significant bacterial pathogens of aquatic animals in the marine environment [1]. It affects both fish [2] and crustaceans [3] and has been related to catastrophic losses in aquaculture in different parts of the world. In shrimp aquaculture, *V. harveyi* is the causative agent of luminous virbiosis [4] but it is also considered one of the causative agents of acute hepatopancreatic necrosis disease (AHPND) [5]. In finfish aquaculture, *V. harveyi* affects many different commercially important fish species, such as groupers (many different species), gilthead seabream (*Sparus aurata*), sole (*Solea* spp.) and European seabass (*Dicentrarchus labrax*). In Mediterranean aquaculture, *V. harveyi* has become a major issue, especially in European seabass aquaculture, in which it causes significant mortality, especially in young individuals when the water temperature is above 20 °C [6]. Currently, there are no commercially available vaccines for vibriosis caused by *V. harveyi*, and the disease is treated with antibiotics such as flumequine, ampicillin and oxytetracycline [7]. Several strains have shown increased resistance to antibiotics and recurrent cycles of different antibiotic treatments are not uncommon. The administration of antibiotics is problematic for the aquaculture industry, as it poses significant threats to its sustainability [8]. Antimicrobial resistance is regarded as the most important threat to humanity, and agriculture and livestock farming including aquaculture are among the major contributors to the problem [9]. Phage therapy has been considered a very promising alternative to antibiotics and is currently being investigated both in human and veterinary applications [10]. Bacteriophages, broadly referred to as phages, are highly specific bacterial viruses that exclusively infect and lyse bacterial hosts. They are the most numerous “life entity” on the planet, occurring naturally in the environment together with their hosts. Lytic phages propagate inside the bacterial cells, which lyse at the end of the propagation cycle, releasing new virions that will continue the infection cycle in the presence of suitable hosts. The targeted killing efficacy of the phages is taken advantage of in phage therapy. Phages against *V. harveyi* have been isolated in the past for strains infecting both shrimps and fish [11,12,13]. Because of their high host specificity, which is usually translated into a narrow spectrum of activity in only few strains of the pathogen, but also the fast emergence of bacterial resistance against phage infection, the efficacy of phage therapy is usually based on the combination of different phages in the form of a cocktail [14]. Therefore, there is an ongoing effort to isolate and characterize novel phages against *V. harveyi*. In this regard, herein, we present the isolation and characterization of a novel phage infecting a *Vibrio harveyi*, which caused vibriosis in a commercial fish farm of European seabass in Greece.

## 2. Results

### 2.1. vB_VhaS_MAG7 Morphology and Characteristics

A novel phage named vB_VhaS_MAG7 was isolated from water samples obtained from a commercial fish farm at the Saronic Gulf in Greece. The novel phage formed clear pinhole plaques in the host loan. Several single plaques were collected and purified through subsequent infections for a total of six times, ensuring each time that plaque morphology was consistent. Observations in TEM revealed that vB_VhaS_MAG7 has a long, no contractile tail with an icosahedral capsid; morphological characteristics classified the novel phage to the siphovirus-like morphotype (Figure 1). The diameter of the capsid was 75 ± 04 nm, the tail was 93 ± 08 nm long and the base plate was 14 ± 04 nm wide. 

### 2.2. Host Range

According to host range analysis, vB_VhaS_MAG7 was found to be a strain-specific phage. No lysis was observed in the bacterial strains that were used in this study except for *V. harveyi* MM46 (Table 1). 

### 2.3. Thermal and pH Stability of vB_VhaS_MAG7

Phage vB_VhaS_MAG7 was exposed to different temperatures to assess its thermal stability. The phage titer remained stable up to 45 °C (F(3, 14) = 1.498, *p* = 0.6392), while a significant decrease was observed at 65 °C compared to the control (F(3, 14) = 7.774, *p* < 0.0001) (Figure 2).

Regarding the behavior of vB_VhaS_MAG7 to various pH, no significant reduction in the phage titer was observed from 4 to 10 pH [F (9, 20) = 1426, *p* = 0.2425]. However, the phage was completely inactivated at pH values of 1, 2 and 3 (Figure 3).

### 2.4. One-Step Growth of vB_VhaS_MAG7

A one-step growth assay revealed that the latent phase for the vB_VhaS_MAG7 was 20 min. The eclipse phase (the time required for the phage to produce the first mature viral particle inside the bacterium) was estimated to be ~10 min, while the burst size was found to be 1393 PFU cell^−1^. vB_VhaS_MAG7 reached the plateau phase at approximately 80 min after the infection (Figure 3). 

### 2.5. In Vitro Cell Lysis

vB_VhaS_MAG7 was able to significantly inhibit the bacterial growth of MM46 up to 33% in MOI 0.1, 1 and 100 compared to the uninfected population (F (4, 670) = 193.3, *p* < 0.0001) after 24 h of incubation (Figure 4) as shown via in vitro analysis. A major decrease in the bacterial population was observed at about 5 h after infection at a multiplicity of infection of 100. 

### 2.6. Genomic Analysis of vB_VhaS_MAG7

Genome sequencing of vB_VhaS_MAG7 produced 14,574,756 raw reads with an average reading length of 150 bp. The phage genome was assembled into a single contig with a minimum coverage of 5× The genome size was 49,315 bp, and the gene distribution was rather dense, as indicated by 1.55 genes per kbp. A total of 76 open reading frames (ORFs) were identified (Table 2), in which 70 (92%) ORFs used ATG as a start codon, 3 ORFs (4%) used GTG and 3 ORFs (4%) used TTG. The vast majority of ORFs were annotated as hypothetical proteins, while 33 ORFs were assigned protein functions based on the similarity with homolog proteins in NCBI and Swissprot databases. No tRNAs were found. Moreover, there were no homologs of integrase, virulence-associated genes or genes encoding antibiotic resistance (Figure 5). Bacphlip analysis showed a 90% probability that vB_VhaS_MAG7 follows a lytic lifestyle. 

Several genes linked to phage packaging and structural assembly were identified, as well as genes that are associated with DNA replication and nucleotide metabolism. No gene clusters or subclusters were observed, indicating that there is no specific genome arrangement. 

Genes related to tail structure and assembly such as tail fiber protein (ORF 30), baseplate protein (ORF 40), putative tail protein (ORF 49), tail assembly chaperone protein (ORF 57), major tail protein (ORF 63), tail protein (ORF 64) and tape measure protein (ORF 55) were identified, as well as genes that encode major structural components such as major capsid protein (ORF 68). Large terminase subunit (ORF 73) and portal protein (ORF 71), which are involved in the packaging process, were also present in the genome. 

Moreover, genes responsible for DNA manipulation and nucleotide metabolism, namely helicase (ORF 5), DNA primase/polymerase (ORF 3), a homing endonuclease (ORF 75), endodeoxyribonuclease (ORF 23), cytosine specific methyltransferase (ORF 48) and DNA binding HTH domain protein (ORF 58), were found. Interestingly, miscellaneous genes such as PD-(D/E)XK nuclease superfamily protein (ORF 7) were also included in the phage genome. 

### 2.7. Genomic Synteny and Phylogenetic Analysis

Phage vB_VhaS_MAG7 belongs to the class of *Caudoviricetes* and is part of a *Siphoviridae* morphology cluster, as shown using proteomic tree analysis (Figure 6) that confirmed the TEM morphological observations. Furthermore, the potential host of MAG7 was found to belong to the taxa of *Gammaproteobacteria*, which includes the *Vibrionaceae* family. Genome comparative analysis among vB_VhaS_MAG7 and other similar phages revealed that vB_VhaS_MAG7 is a novel phage, since the genetic distance between vB_VhaS_MAG7 and the closest phage was 53.9%, which is below the 95% species threshold (Appendix A). 

Phylogenetic analysis using large terminase subunits of the closest phages in Vb_VhaS_MAG7 (Figure 7) showed that MAG7 has a recent common ancestor with Vibrio phage H188, with a bootstrap value of 100%. In addition, the length of the branch indicates that MAG7 has a greater number of amino acid substitutions in its large subunit, as it deviates from its common ancestor with H188. 

### 2.8. In Vivo Phage Therapy in Gilthead Seabream Larvae

In vivo phage treatment trials were carried out in gilthead sea bream larvae to evaluate the effectiveness of vB_VhaS_MAG7 in the control of *Vibrio harveyi* MM46. The bacterial strain *V. harveyi* MM46 was found to be pathogenic, significantly reducing the survival of the larvae to up to 51% compared to the control group, in which 94.6% of the larvae survived during the 5-day trial (X^2^(1, 190) = 46.77, *p* < 0.0001). The survival of gilthead sea bream larvae increased significantly (significance level *p* < 0.05) when treated with vB_VhaS_MAG7 at MOI 10 compared to the group infected with *Vibrio harveyi* MM46 (X^2^(1, 187) = 5.295, *p* = 0.0214). The control group of the phage (without the addition of bacteria) also did not have a significant difference compared to the control group (X^2^(1, 189) = 0.5529, *p* = 0.4571), indicating the safety of the phage suspension and possibly the absence of endotoxins (Figure 8). 

## 3. Discussion

The number of outbreaks caused by *Vibrio harveyi* is on the rise as the effects of climate change become more prominent. This bacterium poses a threat to various marine organisms such as abalones, shrimps, corals, and fishes [15,16,17,18] and can result in major economic losses for the aquaculture industry. Moreover, many *V. harveyi* strains are considered multi-drug resistant, and therefore pose a great challenge. To address this issue, phage therapy has been proposed to control vibriosis. In this context, there has been an increasing number of studies regarding the isolation of phages against *V. harveyi* [13,19,20,21]. In this study, we isolated a new lytic bacteriophage, vB_VhaS_MAG7, which targets *V. harveyi* MM46 and we evaluated its potential as a therapeutic agent. 

As evidenced by its comparison to other phages in the NCBI nr database using a BLAST search and VIRIDIC, vB_VhaS_MAG7 is considered a novel phage. The closest related phage, Vibrio phage H188, which infects *Vibrio kanaloae* [22], was found to have 95% similarity over 48% query cover with MAG7. Although analyzing the evolutionary relationships of phages can be difficult and uncertain due to their wide variety and the mosaic nature of their genomes, phylogenetic analysis based on the signature gene, large terminase subunits, was well supported, further enforcing our findings. Phage vB_VhaS_MAG7 is a *siphovirus,* as shown in TEM. Viral proteomic analysis also showed that the phage was part of a cluster consisting of siphoviruses. vB_VhaS_MAG7 is a tailed phage with a double-stranded DNA genome and therefore is a part of the *Caudoviricetes* class, as of the new taxonomic system [23]. 

Vibrio phages have been linked to inducing virulence in their hosts [24,25]. However, no evidence of virulence factors such as toxins or AMR genes was found in vB_VhaS_MAG7 genome, pointing out its safety as a therapeutic agent. Moreover, no integrases or genes that are associated with genome integration or recombination were present, indicating a vB_VhaS_MAG7 lytic lifestyle. This was further supported by the BACPHLIP analysis. The genome of vB_VhaS_MAG7 contained genes encoding potential structural proteins including capsid protein, major tail protein, and baseplate protein. Genes linked to phage assembly such as large terminase subunit, portal protein and tail assembly chaperone protein were also identified. Interestingly, a gene encoding a cytosine-specific methyltransferase was present. Mono-specific orphan methyltransferases provide protection against digestion by several restriction host’s endonucleases and are considered an important defense mechanism of the phage against the R m modification systems of the bacterial host [26]. Therefore, a more in-depth study is necessary to gain a clearer comprehension of the function of this gene, thereby improving our understanding of phage–host interactions.

Temperature and acidity are crucial factors that play a major role in phage adsorption to the host and phage neutralization, determining the efficacy of phage treatment [27]. Phage vB_VhaS_MAG7 remained consistent in a broad range of temperatures and pH values, which makes it a highly practical candidate for phage therapy. 

As shown in the host range analysis, vB_VhaS_MAG7 appears to be a strain-specific phage, solely infecting *V. harveyi* MM46. Usually, phages with a wide host range are considered ideal for phage therapy, particularly in the field of aquaculture, where pathogenic strains and species exhibit a high degree of diversity. However, in this case, we consider vB_VhaS_MAG7 an excellent candidate due to its efficient lytic ability both in vitro and in vivo. Remarkably, vB_VhaS_MAG7 burst size was unusually large. A few phages have been previously reported [13,28,29,30] to produce a substantial number of virions, including Vibrio phage Virtus, which also targets *V. harveyi.* Burst size is a vital characteristic for a phage and it is affected by a number of elements, such as the metabolic activity of the host bacteria, the surrounding environment, and the host’s protein synthesis system [28,31,32]. That said, further investigation is required to comprehend the molecular mechanism associated with its remarkable burst size. 

Multiple studies have shown that phages are effective at treating vibriosis in various animal models [11,21,33]. For instance, the survival of abalone was improved by 70% in 7 days as shown by Wang et al. [33], and Vinod et al. showed a significant increase in the survival of giant tiger prawn (*Penaeus monodon*) treated with phages [11]. Moreover, we have previously demonstrated a successful phage therapy scheme whereby Vibrio phage Virtus [13] was able to increase the survival of gilthead seabream larvae by 35% compared to the untreated population. Here, gilthead seabream larvae that were treated with vB_VhaS_MAG7 had 19% higher probability of survival in comparison with the untreated group. In addition, the phage control group showed no difference in terms of survival compared to the control group, corroborating vB_VhaS_MAG7 safety as a therapeutic agent. 

In conclusion, Vb_VhaS_MAG7 is a novel strain-specific phage against *V. harveyi.* The distinct biological and genetic features of Vb_VhaS_MAG7, as well as its efficacy as a therapeutic agent, make it a suitable candidate for phage therapy as a safe and effective option. 

## 4. Materials and Methods

### 4.1. Bacterial Strains

Bacterial strains that were used in this study (Table 1) belong to the bacterial collection of Laboratory of Aquaculture Microbiology, Institute of Marine Biology, Biotechnology and Aquaculture (IMBBC), Hellenic Center for Marine Research (HCMR) in Heraklion, Crete. They were identified at the species level either through whole-genome sequencing (WGS) and/or through sequencing of 16s rRNA and mreB genes and PCR detection of toxR, as described in Droubogiannis et al. [34]. *Vibrio harveyi* strain MM46 which had been isolated from diseased seabass from the same fish farm was used as a host for phage isolation. Identification of the host was performed following whole-genome sequencing (NZ_JAPPTO000000000.1). All bacterial strains were maintained in microbeads (MicroBank, Pro-Lab Diagnostics, Ontario, Canada) at −80 °C and were grown in lysogeny broth (10 gL^−1^ tryptone, 5 gL^−1^ NaCl, 1 L deionized water, 0.75 gL^−1^ MgSO_4_, 1.5 gL^−1^ KCl, 0.73 gL^−1^ CaCl_2_) at 25 °C prior to the experiments.

### 4.2. Antibiotic Susceptibility Testing 

The bacterial strains used in this study were tested for antibiotic susceptibility using a standard disk diffusion test [35]. To carry out this test, bacterial suspensions of the *V. harveyi* MM46 were diluted to obtain a specific absorbance reading at OD_600_. The diluted bacterial suspensions were then plated on Mueller–Hinton agar (a type of nutrient agar) with 2% NaCl. Antimicrobial susceptibility disks (provided by ThermoFisher Scientific, Waltham, Massachusetts, United States) were placed on the agar plates, and the plates were incubated at 25 °C (which is the optimal temperature for the bacteria used) for 24 h. The diameters of the zones of inhibition around the disks were measured and recorded, and these measurements were interpreted as susceptible, medium, or resistant according to the Clinical Laboratory Standards Institute (CLSI) guidelines (specifically, CLSI M45-A2 [36] and CLSI M100-S25 [37]). Table 3 shows the details of the antimicrobial susceptibility disks used in the study and the corresponding interpretations of the recorded diameters.

### 4.3. Isolation and Purification of Bacteriophages

The bacteriophage vB_Vh_MAG7 was isolated from water samples taken from a commercial fish farm in Greece. Grab sampling was used to collect water using a sterile container which was then transported at low temperature—to prevent the growth of any phages present in the sample—to the laboratory for analysis. The standard isolation and phage purification procedures were followed according to Clokie et al. [38], as described by Droubogiannis et al. [13]. Specifically, the enrichments underwent centrifugation at a speed of 13,000 rpm for a duration of 3 min, and the resulting supernatants were then passed through a sterile filter with a pore size of 0.22 µm (manufactured by GVS Life Sciences located in Sanford, ME, USA). Next, 10 µL of each sample was placed on bacterial lawns of the host strain, and the clearest plaques were collected after a 24 h incubation period. These plaques were subsequently propagated against the host strain using the double agar layer method, as originally described by Clokie et al [38] and reported in the work of Droubogiannis et al. [13]. This entire procedure was repeated five times before the phage was considered to be purified. The purified phage was maintained either in lysogeny broth (10 gL^−1^ tryptone, 5 gL^−1^ NaCl, 1 L deionized water, 0.75 gL^−1^ MgSO_4_, 1.5 gL^−1^ KCl, 0.73 gL^−1^ CaCl_2_) or in SM buffer (5.8 gL^−1^ NaCl, 2 gL^−1^ MgSO_4_, 50 mL Tris-Cl (1 M, pH 7.5)) at 4 °C. Prior to the experiments, we determined the titer of the phage using a double agar assay, as described in Clokie et al. [38].

### 4.4. Transmission Electron Microscopy

Transmission Electron Microscopy was conducted at the Electron Microscopy Lab of the University of Crete using a JEOL JEM-2100 (JEOL Ltd., Akishima, Tokyo, Japan) transmission electron microscope in order to observe phage morphology. Phage sample preparation included negative staining with 4% *w*/*v* uranyl acetate, 7.2 pH as previously described in Misol et al. [21]. Morphological characteristics of the phage were analyzed and measured using ImageJ (software version 1.53p). A total of n = 30 measurements was performed. 

### 4.5. Host Range Assay

Several bacterial strains were used to evaluate the host range of the phage (Table 1). Briefly, 1 mL of a fresh broth bacterial culture (~10^7^ CFU mL^−1^) was added to 3 mL of LB soft agar (0.75% agar) following the solidification of the mixture. A spot test was then performed where 10 μL of the phage dilutions (initial titer ~10^9^ PFU mL^−1^) were spotted in the bacterial lawn of each individual bacterial strain in order to evaluate the efficacy of the phage. Phage plaques were counted the following day and the titer was calculated as PFU mL^−1^.

### 4.6. Stability of Phage in Different Temperatures and pH Values

The thermal stability of the novel phage was examined as described in Droubogiannis et al. [13] by exposing aliquots of the phage (titer: 10^7^ PFU mL^−1^) to various temperatures (4, 25, 35, 45, 65, 70 and 80 °C). The aliquots were incubated at each temperature for 1 h and then rested at room temperature (RT) for 10 min. Each aliquot was then serially diluted and spotted (10 µL/spot) on a host bacterial lawn. After a 24 h incubation of the agar plates, the phage titer was determined for each temperature and 4 °C was used as the control.

The stability of phage titer at various pH levels was assessed as described by Pan et al. [28]. The desired pH values, ranging from 1 to 10, were achieved by adding NaOH or HCl to LB broth following the suspension of phages with a final titer of ~10^−7^ PFU mL^−1^. The suspensions were stored at 4 °C for 2 h. After a 10 min rest to allow the suspension to adjust to room temperature (RT), aliquots were taken and serially diluted and spotted onto the host bacterial lawn. After a 24 h incubation of the agar plates, the phage titer was determined for each pH value and pH = 7 was used as the control. Both the thermal and pH stability was tested in triplicate.

### 4.7. One-Step Growth Assay

A one-step growth assay was conducted as described in Droubogiannis et al. [34]. In brief, 1 mL of a fresh culture (10^7^ CFU mL^−1^) was centrifuged at 13,000 rpm for 3 min. The supernatant was discarded and the pellet was saline washed. The procedure was repeated twice, and the pellet was resuspended in LB, infected with the phage at a MOI of 0.01 and rested for 15 min at room temperature. The supernatant was removed after 3 min of centrifugation at 13,000 rpm, and the pellet was dissolved in 1 mL saline. The infected culture was then transferred to a fresh tube containing 25 mL LB, and aliquots were taken and deposited in empty Eppendorf vials. The aliquots were then centrifuged at 13,000 rpm for 3 min and the supernatant was transferred to a new 2 mL Eppendorf vial, serially diluted, and spotted onto the host bacterial lawn on LB ½ agar plates. This procedure was repeated every 10 min for 120 min. The phage titer was determined after 24 h of incubation of the agar plates at 25 °C. The burst size was calculated as the ratio of the final count of released virions at the end of the burst period to the initial count of infected bacterial cells at the beginning of the latent period.

### 4.8. In Vitro Cell Lysis

To assess the effectiveness of vB_VhaS_MAG7 in vitro against *V. harveyi* MM46, 180 µL of fresh host bacterial culture (~10^6^ CFU mL^−1^) was loaded into each well of sterile 96-well plates. The plates were then read at OD_600_ using a TECAN microplate reader (Infinite PRO 200) with orbital shaking at 25 °C. When the host culture was in the exponential phase (≈10^7^ CFU mL^−1^), vB_VhaS_MAG7 was added at MOIs of 0.1, 1, 10, and 100. Two control treatments were also used in triplicate; one negative control with only LB and one positive control where *V. harveyi* MM46 population was added. The growth curves of the cultures were measured every 10 min for 24 h, and all assays were performed in triplicate.

### 4.9. Genomic Analysis

The genetic material of vB_Vh_MAG7 was extracted using the phenol–chloroform method, as described by Higuera et al. [39]. At least 5 µg of high-purity bacterial DNA was used to generate a paired-end 300 PE genomic library using Nextera library preparation kit (Illumina, San Diego, CA, USA). The quality of DNA was tested using a BioAnalyzer (Bio-Rad, Chicago, IL, USA), as described previously [14]. Sequencing was performed using an Illumina NovaSeq 6000 sequencing platform (Illumina, San Diego, CA, USA) according to the manufacturer’s protocol. Possibly contaminated, primer, N-terminus and 30- or 50-low quality reads were trimmed off (threshold: 0.05). 

The Geneious assembler was used for the assembly of the paired reads using Geneious Prime software. The genome annotation pipeline, previously described by Droubogiannis et al. [34], was followed. Both structural and functional annotation were conducted in the Galaxy webserver [40] environment. Predicted ORFs were manually checked for the presence of a valid start and stop codon and a Shine-Dalgarno sequence. Several databases, including NCBI Basic Local Alignment Search Tool (BLAST) adjusted at a non-redundant (nr) protein database [41], Swissprot, Gene Ontology [42], Interpro [43] and TMHMM 2.0 were used to assess protein homology for functional annotation. Genes associated with integration, virulence and antibiotic resistance in the phage genome were searched for using the INTEGRALL Database webserver [44] and Virulence Factor Database (VFDB) [45], as well as the VirulenceFinder and ResFinder webservers [46]. The phage lifestyle was predicted using BACPHLIP through Galaxy Webserver [47]. The genome of vB_VhaS_MAG7 with annotated predicted ORFs was then visualized as a circular representation with Geneious software (Geneious v9.1, Biomatters, Auckland, New Zealand). Comparative phage genome analysis was conducted using VIRIDIC, which calculates intergenomic similarities between viral genomes [48].

### 4.10. In Vivo Phage Therapy Trial in Gilthead Seabream Larvae

In vivo phage therapy trials were conducted according to Droubogiannis et al. [13]. Gilthead seabream eggs at the same developmental stage were obtained from HCMR hatchery, washed three times with sterile sea water, and placed individually in a 96-well microplate (1 egg/well) containing 200 μL sterile sea water. After one day of incubation, the quality of eggs was evaluated according to Panini et al. [49]. The challenge test started when larvae were hatched.

Bacteria used in the challenge test were cultivated in LB overnight and diluted 1:100 in fresh LB. After a 2 h incubation at 25 °C, cells were centrifuged and washed twice with buffer A (saline 0.9%, MgCl_2_ 10 mM). The bacterial suspensions were adjusted to ~10^7^ CFU mL^−1^ with buffer A. No treatment was performed in the first group of larvae. The second group was treated with vB_VhaS_MAG7 alone (without addition of bacteria) at an approximate concentration of 10^8^ PFU mL^−1^ and served as a negative phage control. The third group was treated with 10^6^ CFU mL^−1^ *Vibrio harveyi* MM46. Finally, the fourth group was treated with 10^6^ CFU mL^−1^ *Vibrio harveyi* MM46 and vB_VhaS_MAG7 at 10 ΜOΙ. Phage suspensions were treated with 10% (*w*/*v*) PEG overnight at 4 °C to remove possible endotoxins in the phage lysate and diluted in SM buffer (NaCl 100 mM, MgSO_4_7H_2_O 8 mM, Tris-Cl 1 M; pH 7.5). The phage titer was also determined prior to the experiment with a double agar assay. Phage suspensions were added to the corresponding treatments two hours after infection. In addition, all controls were treated the same way, but instead of phage lysate, saline 0.9% was added to each well. The survival of gilthead seabream larvae was monitored daily for the following five days. 

### 4.11. Statistical Analysis

One-way ANOVA was performed for the thermal, pH stability and in vitro assays along with Dunnett’s multiple comparison test [50]. Tukey’s HSD post hoc test [51] was used as a multiple comparison tool after ANOVA was performed. Kaplan–Meier survival analysis [52] was performed for the in vivo phage therapy trial in gilthead seabream larvae. All statistical analyses were carried out using GraphPad Prism version 9.0.0 for Windows, GraphPad Software, San Diego, CA, USA).

## Figures and Tables

**Figure 1 ijms-24-08200-f001:**
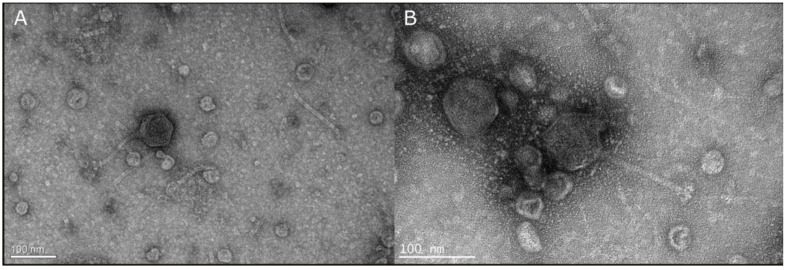
(**A**) Transmission electron micrograph of vB_VhaS_MAG7 showing the typical morphology of the Siphovirus morphology group. (**B**) Higher magnification of the phage showing details of the tail.

**Figure 2 ijms-24-08200-f002:**
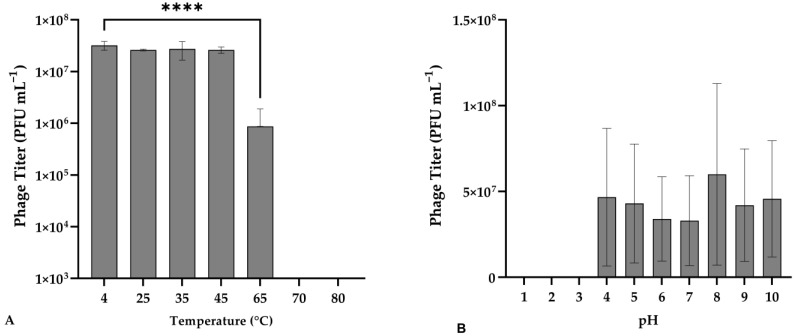
(**A**) Effect of different temperatures on the stability of vB_VhaS_MAG7. Incubation at 4 °C for 1 h was used as control. (**B**) Effect of pH on the stability of vB_VhaS_MAG7. Incubation with pH = 7 for 2 h was used as control. Phage titer was measured against *V. harveyi* MM46. Error bars were shown for the mean of n = 3. Statistical significance indicated by **** at *p* < 0.0001.

**Figure 3 ijms-24-08200-f003:**
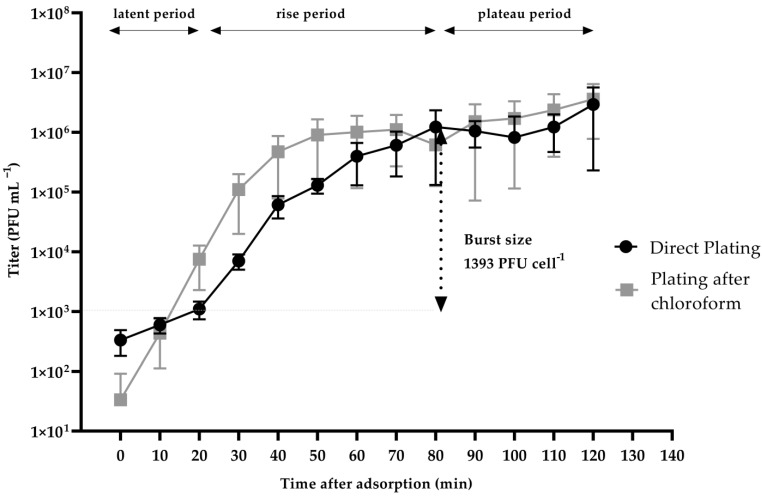
One-step growth of vB_VhaS_MAG7 measured against *V. harveyi* MM46 at multiplicity of infection (MOI) 0.01. Error bars were shown for the mean of n = 3.

**Figure 4 ijms-24-08200-f004:**
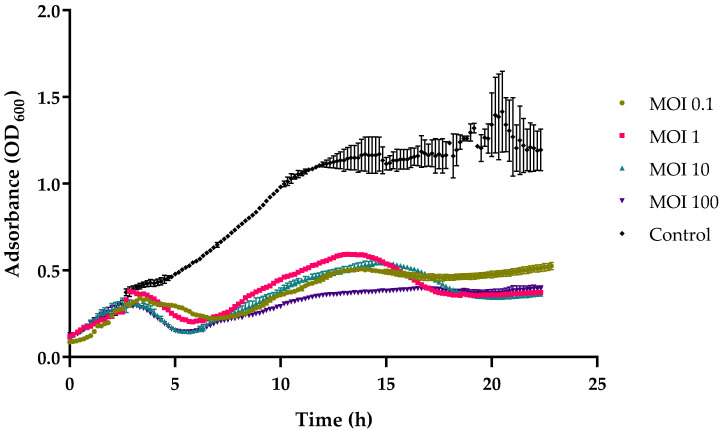
In vitro lysis of vB_VhaS_MAG7 against *V. harveyi* MM46 at MOIs 0.1, 1, 10, and 100 for 24 h. Bacterial growth indicated by the absorbance (OD_600_) read. Error bars were shown for the mean of n = 3.

**Figure 5 ijms-24-08200-f005:**
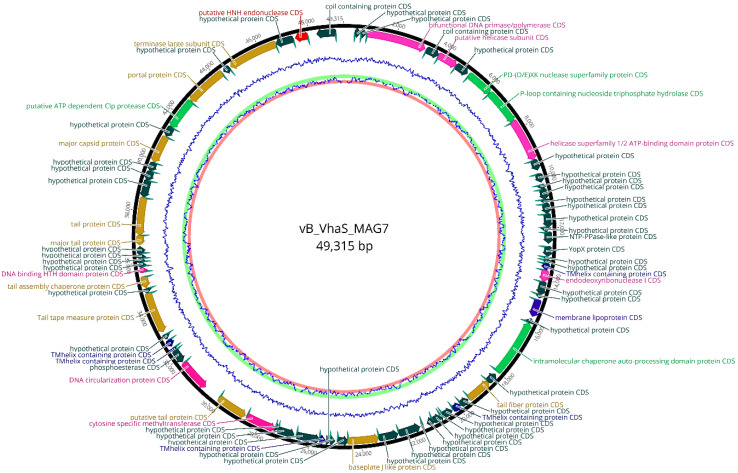
Visual representation of vB_VhaS_MAG7 genome. The GC content of the genome is shown by the inner blue line. Predicted ORFs are shown in the outer circle. ORF color refers to annotated biochemical function: phage assembly proteins (brown), DNA replication proteins, repair and recombination (purple), miscellaneous proteins (light green), transmembrane proteins (blue), hypothetical proteins (dark green), Shine Dalgarno sequences (cyan).

**Figure 6 ijms-24-08200-f006:**
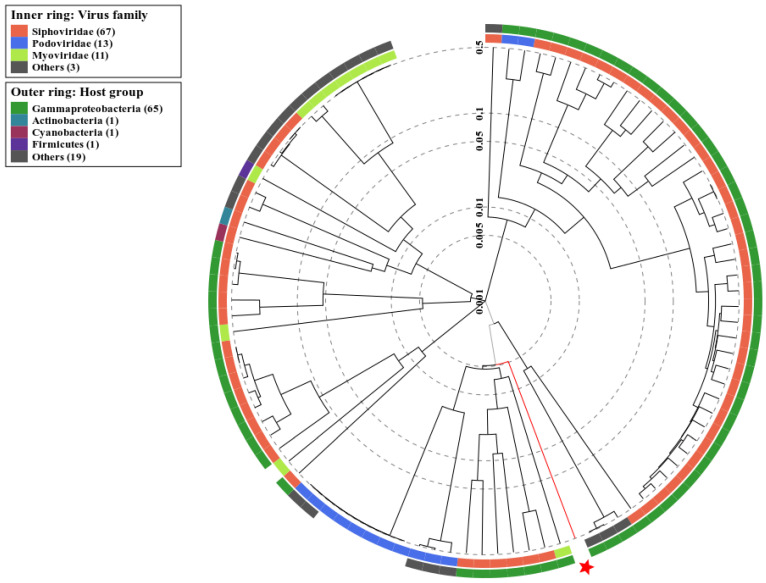
Identification of classes and host group for Vb_VhaS_MAG7 according to the proteomic tree produced by VIPTree. Vb_VhaS_MAG7 belongs to a *Siphoviridae* cluster and infects hosts from the *Gammaproteobacteria* group (red star and line). The branch length scale was calculated as log values. The inner and outer rings indicate the virus taxonomic family and the host group, respectively.

**Figure 7 ijms-24-08200-f007:**
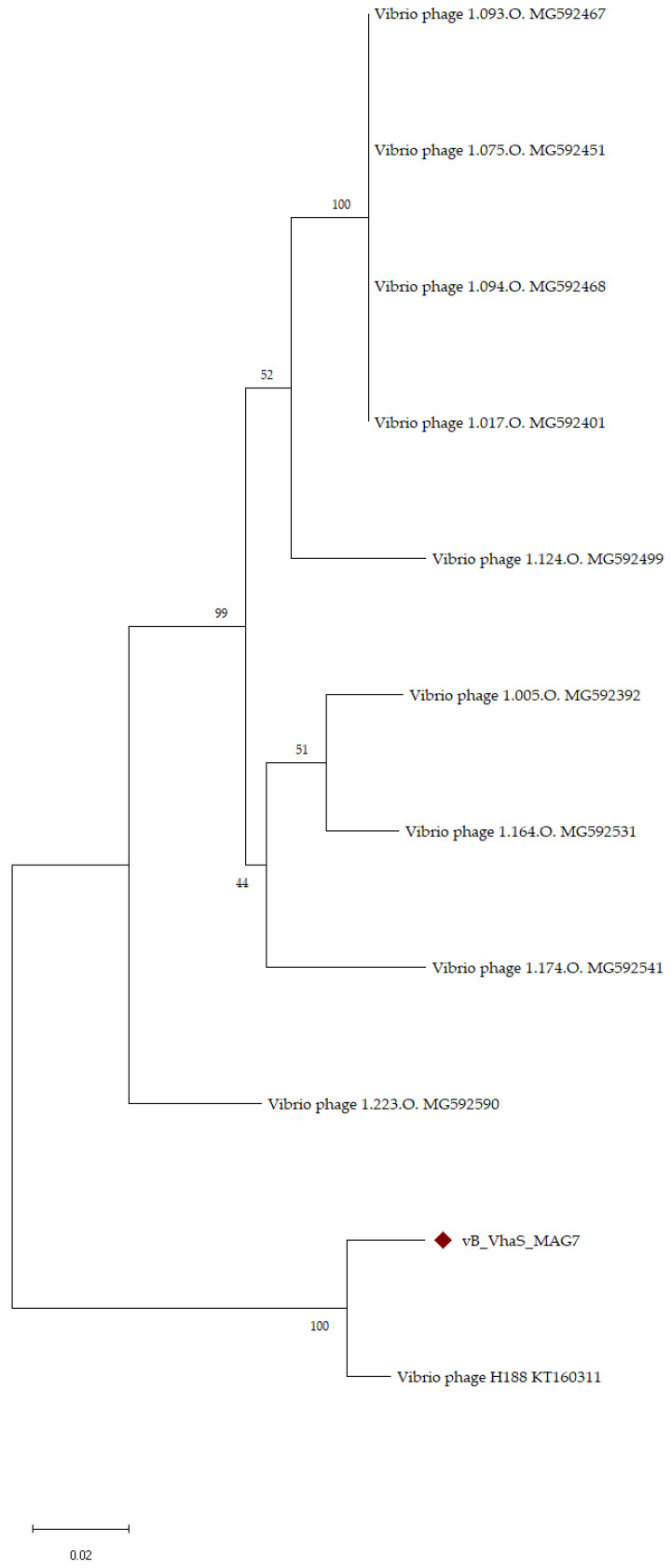
Phylogenetic tree of vB_VhaS_MAG7 and closest phages based on the large terminase subunits. Large phage terminase subunits were downloaded from the NCBI database and aligned using MUSCLE. A rooted phylogenetic tree of maximum probability (bootstrap = 1000) was constructed using MEGA X. The phages with the largest genomic distance from vB_VhaS_MAG7 were arbitrarily used as an outgroup. The “diamond” symbol represents vB_VhaS_MAG7.

**Figure 8 ijms-24-08200-f008:**
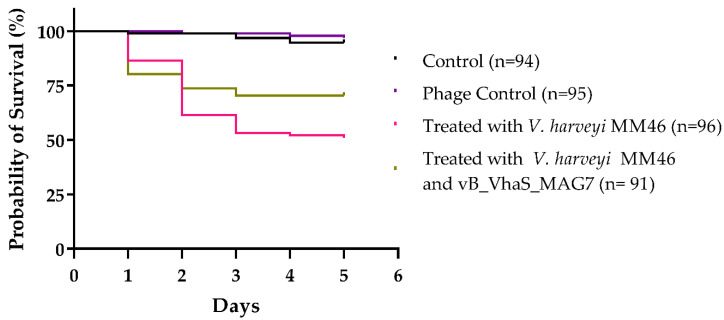
Survival of gilthead seabream larvae infected with *V. harveyi* MM46 in an experimental phage treatment trial for 5 days. Gilthead seabream larvae infected with MM46 (~10^7^ CFU mL^−1^) were treated with vB_VhaS_MAG7 at MOI 10, two hours after infection. Statistical significance at *p* < 0.0001.

**Table 1 ijms-24-08200-t001:** Host range of vB_VhaS_MAG7 against *Vibrio* spp.

Strain	Species	Country	Host	Lysis
MM46 *	*V. harveyi*	Greece	*Sparus aurata*	+
DSM 19623	*V. harveyi*	USA	*Talochestria capensis*	-
SA 2.1	*V. harveyi*	Saudi Arabia	*Sparus aurata*	-
DSM 2171	*V. alginolyticus*	Japan	*Trachurus trachurus*	-
Gal 90	*V. harveyi*	Greece	*Sparus aurata*	-
Vh No22	*V. harveyi*	Greece	*Dicentrarchus labrax*	-
Kef 62	*V. harveyi*	Greece	*Dicentrarchus labrax*	-
Kef 75	*V. harveyi*	Greece	*Dicentrarchus labrax*	-
Gal 56	*V. harveyi*	Greece	*Dicentrarchus labrax*	-
Gal 77	*V. harveyi*	Greece	*Sparus aurata*	-
Gal 72	*V. harveyi*	Greece	*Dicentrarchus labrax*	-
Gal 94	*V. harveyi*	Greece	*Sparus aurata*	-
L. SUSI	*V. parahaemolyticus*	Philippines	Shrimp	-
V1	*V. alginolyticus*	Greece	*Sparus aurata*	-
LAR194	*V. mediterranei*	Greece	*Artemia* nauplii	-
SM1	*V. harveyi*	Greece	*Seriola dumerili*	-
MAN113	*V. splendidus*	Greece	*Seriola dumerili*	-
VarvA4 1.1	*V. harveyi*	Greece	*Sparus aurata*	-
VH2	*V. harveyi*	Greece	*Seriola dumerili*	-
VhP1 Liv	*V. harveyi*	Greece	*Seriola dumerili*	-
VhP1 Spl	*V. harveyi*	Greece	*Dicentrarchus labrax*	-
DY05	*V. owensii*	Greece	*Dicentrarchus labrax*	-
SA 6.2	*V. owensii*	Saudi Arabia	*Oreochromis niloticus*	-
VIB391	*V. campbellii*	Thailand	Shrimp	-
Kef 56	*V. rotiferianus*	Greece	*Dicentrarchus labrax*	-
VhSerFre	*V. harveyi*	Greece	*Seriola dumerili*	-
sngr	*V. harveyi*	Greece	*Dentex dentex*	-
ks6	*V. owensii*	Greece	*Dicentrarchus labrax*	-
VH5	*V. harveyi*	Greece	*Seriola dumerili*	-
RG1	*V. harveyi*	Greece	*Dicentrarchus labrax*	-
Serkid	*V. harveyi*	Greece	*Seriola dumerili*	-
SERKID2	*V. harveyi*	Greece	*Seriola dumerili*	-
SERSD	*V. harveyi*	Greece	*Seriola dumerili*	-
SA 5.1	*V. harveyi*	Saudi Arabia	*Sparus aurata*	-
SA 6.1	*V. harveyi*	Saudi Arabia	*Sparus aurata*	-
SA 9.2	*V. harveyi*	Saudi Arabia	*Sparus aurata*	-
SA 1.2	*V. harveyi*	Saudi Arabia	*Sparus aurata*	-
SA 7.1	*V. harveyi*	Saudi Arabia	*Sparus aurata*	-
SA 3.1	*V. harveyi*	Saudi Arabia	*Sparus aurata*	-
SA 4.1	*V. harveyi*	Saudi Arabia	*Sparus aurata*	-
SA 2.1	*V. harveyi*	Saudi Arabia	*Sparus aurata*	-
VH6	*V. harveyi*	Greece	*Dicentrarchus labrax*	-
V2	*V. alginolyticus*	Greece	*Dentex dentex*	-
SA 1.1	*V. owensii*	Saudi Arabia	*Sparus aurata*	-
SA 9.1	*V. owensii*	Saudi Arabia	*Sparus aurata*	-

*: host strain, +: lysis, -: no lysis

**Table 2 ijms-24-08200-t002:** Summary table of vB_VhaS_MAG7 genes annotated with relevant information based on important amino acid sequences and structural protein homologations (E-value ≤ 10^−3^).

Type	Predicted Functions	Start	End	Length	Strand
ORF 1	Hypothetical protein	401	629	72	Forward
ORF 2	Hypothetical protein	633	868	74	Forward
ORF 3	Bifunctional DNA primase/polymerase	871	3183	767	Forward
ORF 4	Coil-containing protein	3237	3762	170	Forward
ORF 5	Putative helicase subunit	3762	4551	259	Forward
ORF 6	Hypothetical protein	4542	5095	180	Forward
ORF 7	PD-(D/E)XK nuclease superfamily protein	5151	6229	355	Forward
ORF 8	P-loop-containing nucleoside triphosphate hydrolase	6212	7641	471	Forward
ORF 9	Helicase superfamily 1/2 ATP-binding domain protein	7611	9317	563	Forward
ORF 10	Hypothetical protein	9307	9739	138	Forward
ORF 11	Hypothetical protein	9870	10,209	108	Forward
ORF 12	Hypothetical protein	10,308	10,496	59	Forward
ORF 13	Hypothetical protein	10,481	10,903	136	Forward
ORF 14	Hypothetical protein	10,889	11,070	56	Forward
ORF 15	Hypothetical protein	11,055	11,462	131	Forward
ORF 16	Hypothetical protein	11,449	11,992	176	Forward
ORF 17	Hypothetical protein	11,959	12,164	65	Forward
ORF 18	NTP-ppase-like protein	12,158	12,601	144	Forward
ORF 19	Yopx protein	12,699	13,096	128	Forward
ORF 20	Hypothetical protein	13,112	13,233	36	Forward
ORF 21	Hypothetical protein	13,220	13,370	45	Forward
ORF 22	Tmhelix-containing protein	13,358	13,627	86	Forward
ORF 23	Endodeoxyribonuclease I	13,614	14,082	152	Forward
ORF 24	Hypothetical protein	14,070	14,594	169	Forward
ORF 25	Hypothetical Protein	14,582	14,737	47	Forward
ORF 26	Membrane lipoprotein	14,795	15,477	223	Forward
ORF 27	Hypothetical Protein	15,565	15,765	63	Reverse
ORF 28	Intramolecular chaperone auto-processing domain protein	15,769	18,027	750	Reverse
ORF 29	Hypothetical protein	18,117	18,535	135	Reverse
ORF 30	Tail fiber protein	18,526	19,525	329	Reverse
ORF 31	Hypothetical protein	19,566	19,908	110	Reverse
ORF 32	Tmhelix-containing protein	19,910	20,263	113	Reverse
ORF 33	Hypothetical protein	20,311	20,720	130	Reverse
ORF 34	Hypothetical protein	20,700	20,813	34	Reverse
ORF 35	Hypothetical protein	20,804	21,118	104	Reverse
ORF 36	Hypothetical protein	21,129	21,385	80	Forward
ORF 37	Hypothetical protein	21,542	21,658	34	Forward
ORF 38	Hypothetical protein	21,688	22,548	282	Reverse
ORF 39	Hypothetical protein	22,567	23,346	255	Reverse
ORF 40	Baseplate J-like protein	23,334	24,534	397	Reverse
ORF 41	Hypothetical protein	24,520	24,943	137	Reverse
ORF 42	Hypothetical protein	25,020	25,217	62	Reverse
ORF 43	Hypothetical protein	25,205	25,360	51	Reverse
ORF 44	Tmhelix-containing protein	25,390	25,630	75	Reverse
ORF 45	Hypothetical protein	25,617	26,541	303	Reverse
ORF 46	Hypothetical protein	26,612	27,280	218	Reverse
ORF 47	Hypothetical Protein	27,243	27,408	50	Reverse
ORF 48	Cytosine specific methyltransferase	27,395	28,586	392	Reverse
ORF 49	Putative tail protein	28,637	29,857	403	Reverse
ORF 50	DNA circularization protein	30,519	31,778	415	Reverse
ORF 51	Phosphoesterase	31,837	32,371	175	Reverse
ORF 52	Tmhelix-containing protein	32,364	32,485	36	Reverse
ORF 53	Tmhelix-containing protein	32,550	32,842	90	Reverse
ORF 54	Hypothetical Protein	32,876	33,085	65	Reverse
ORF 55	Tail tape measure protein	33,191	34,803	533	Reverse
ORF 56	Hypothetical protein	34,796	34,994	60	Reverse
ORF 57	Tail assembly chaperone protein	35,029	35,467	141	Reverse
ORF 58	DNA binding HTH domain protein	35,609	35,835	71	Forward
ORF 59	Hypothetical protein	35,826	36,023	61	Forward
ORF 60	Hypothetical protein	36,005	36,180	54	Forward
ORF 61	Hypothetical protein	36,153	36,356	64	Forward
ORF 62	Hypothetical protein	36,340	36,634	93	Forward
ORF 63	Major tail protein	36,673	37,064	125	Reverse
ORF 64	Tail protein	37,052	38,522	486	Reverse
ORF 65	Hypothetical protein	38,524	39,202	223	Reverse
ORF 66	Hypothetical protein	39,185	39,580	127	Reverse
ORF 67	Hypothetical protein	39,568	39,932	117	Reverse
ORF 68	Major capsid protein	39,934	41,028	360	Reverse
ORF 69	Hypothetical protein	41,031	41,455	136	Reverse
ORF 70	Putative ATP dependent Clp protease	41,441	42,692	412	Reverse
ORF 71	Portal protein	42,679	44,325	543	Reverse
ORF 72	Hypothetical protein	44,386	44,657	85	Reverse
ORF 73	Terminase large subunit	44,708	46,677	652	Reverse
ORF 74	Hypothetical protein	46,666	47,374	232	Reverse
ORF 75	Putative HNH endonuclease	47,435	47,948	167	Reverse
ORF 76	Coil-containing protein	48,283	49,029	245	Reverse

**Table 3 ijms-24-08200-t003:** Antibiotic Susceptibility Testing of *V. harveyi* MM46.

Antimicrobial Agent		Zone Diameter (mm)	Interpretation
Flumequine	UB	30	S
Tetracycline	TE	25	S
Florfenicol	FFC	25	S
Oxytetracycline	OT	26	I
Oxolinic acid	OA	22	S
Trimethoprim/sulfamethoxazole	SXT	15	I
Ampicillin	AMP	-	R
Piperacillin	PRL	-	R

Abbreviations: S, sensitive; I, medium; R, resistant.

## Data Availability

The genome sequence of phage vB_VhaS_MAG7 is available in GenBank under accession number OK381870. The genome sequence of *V. harveyi* MM46 is available in GenBank under accession number NZ_JAPPTO000000000.1.

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
