# Peer review of "Comprehensive Characterization of a Novel Bacteriophage, vB_VhaS_MAG7 against a Fish Pathogenic Strain of Vibrio harveyi and Its In Vivo Efficacy in Phage Therapy Trials"

_ijms, 2023, doi:10.3390/ijms24098200_

Round 1

Reviewer 1 Report

"Comprehensive characterization of a novel bacteriophage, vB_VhaS_MAG7 against a fish pathogenic strain of Vibrio harveyi and its in vivo efficacy in phage therapy trials" from Droubogiannis and Colleges includes the characterization of the phage vB_VhaS_MAG7 with respect to classical parameters such as morphology, host range, stability, growth parameters, efficiency in vitro and in vivo, and genomic analysis. Based on the results, the authors conclude the safe use of the phage in the occurrence of Vivrio harveyi in aquaculture.

The structure of the manuscript and the methods used for the purpose make a solid impression.

However, since very similar publications have already appeared by the working group, it makes very much the impression of copying the basic idea and replacing it with appropriate parameters. I would have liked to see more emphasis on the special features of the phage.

As long as the safety of the phage is proven by recognized methods, there is nothing against the use of this phage for therapeutic purposes, however, this phage seems to be so specific that an economical stocking is not given.

Basically, it would have been nice if a little more attention had been paid to the form before uploading. Tens of different fonts. Not every sentence is ended by a punctuation mark, sometimes twice for that. Spaces should be placed before units. The spelling of the phage varies from vB_VhaS_MAG7 (e.g. line 18) to vB_Vhs-MAG7 (line 19) to Vb_VhaS_MAG7 (line 256) or vB_Vh_MAG7 (line 59). This makes a not really good impression. Formatting in italics is missing in places. Also the wrong pagination on page 6 could have been noticed.

Even though there is little that is "spectacular", there are few rough criticisms included and accordingly considered relevant to the field.

Major comments

One criticism, however, is that the phage genome is not yet available. In my opinion, this is a requirement for this publication. If the sequence is not published, all data based on it should be removed from the publication.

And the in vitro assay is not described in the methods.

Minor comments:

3.1 Strain MM46: how relevant is this strain or how representative if used here as a model pathogen. Since antibiotic resistance is brought into play, it would be desirable if the pathogen was analyzed in this regard. The strain has reportedly been sequenced, but no data are deposited. Unpublished data should be avoided, therefore it would be desirable if the sequence of the strain is published.

3.3 TEM: How many measurements have been performed?

3.4 Host range: Soft agar concentration? Were dilutions spotted?

3.5 Stability: what were the temperatures above 40 degrees and extreme ph-values tested for, are they relevant to phage use???? At the temperature variation, incubation was for 1 h at pH variation 2 h. What is the rationale for this? Is biological or technical replicates meant by replicates?

3.6 One-Step: in the execution, reference is made to another own publication. Also there it does not say how the burst size was calculated concretely. In the text it says later PFU / infected cell. How was the number of infected cells determined? Or indicate how it was actually calculated.

3.9. in vivo: Why does phage control have a different phage concentration (1E8 PFU/ml) than phage treatment (MOI 10, at 1E6 CFU = 1E7 PFU)? This does not make sense.

Line 129: after hatching (eggs cannot hatch).

Missing description of how the phage used was prepared and purified.

3.10 Statistics: list everything here (partly duplicated to the previous section, some evaluations are missing, like for in vitro efficiency)

4.1: Taxonomy outdated, siphoviridae no longer exist. If necessary, refer to as siphovirus morphology. Base plate not recognizable on Fig 1. Add other figure, add number of measurements for determination of dimensions.

Figure 1: Expression Morpho à morphology

Table 1 What does ++++ mean for lysis behavior, include in legend.

Combine Figure 2 and 3, scaling of y-axes should be identical. Label Y-axis: there is only the unit so far. Titer (PFU/mL) or similar would be useful. Indicate the incubation period in the legend.

4.4 One-Step/Figure 4: in the 20 min latency the previous incubation for 15 min was not considered. Therefore, the labeling of the x-axis (time after infection) does not make sense. In the fig the error bars are missing, did the arrow for burst size slip? If not, I don't understand what was calculated.

4.5. How was the significance calculated? No indication in the material and method part, reference to fig incomplete, after 5h the lowest value should be seen at MOI 100. I don't see any difference to MOI 10. how to explain that there is almost no difference between the different MOIs and that there are 2 lysis cycles 1. completed after about 5h, 2. after about 18h?

Figure 7: taxonomy update. I can't see a clear assignment to the siphoviruses in this figure either... Which phages were used for the tree?

Figure 8: which phages are outgroupe?

4.8, Fig 9 and Table 4: Table 4 is redundant. Values can be given in the text or directly in the Fig.;Indicate how many larvae per group; Indicate significance in Fig, indicate CFU in legend.

Line 273: Why was endotoxin level not measured. The conclusion on endotoxin does not make sense here or were cytokines or similar measured to support this.

Discussion:

Line 287 gram- write out

Line 298 Siphoviridae out of date

Line 301 virulence factors

line 307 cytosines?

line 308 sentence structure

Line 311 do the extreme conditions tested play a role in the intended therapy? If so, where? What temperature is normal for aquaculture

In the in vivo experiments, MOI 10 was used to infect. How high is the bacterial load in the field and would such a high dose then be feasible at all? Especially since even at this concentration there would still be 30% losses. Would the use be economical at all, if additional therapy would still be necessary? It would be nice if an outlook could be provided on how the remaining losses could be avoided.

Contribution of LP?

Reviewer 2 Report

The research article by Droubogiannis et al. on "Comprehensive characterization of a novel bacteriophage, vB_VhaS_MAG7 against a fish pathogenic strain of Vibrio harveyi and its in vivo efficacy in phage therapy trials" describes the morphological and genomic characters of a Vibrio phage infecting V. harveyi. There are some major issues in the methods section and the genome annotation.

1. Line no. 31: Name the antibiotics and add references.

2. Line no. 35: Introduce bacteriophages and explain the use of phages as antibacterials with references.

3. Line no. 37-38: Rephrase this sentence.

4. Line no. 45-46: The aim of this study needs to be clearly explained.

5. Line no. 51-52: Provide accession numbers/ genome sequences.

6. Line no. 59: Sample collection should be described in detail, especially for others to repeat a similar study.

7. Line no. 63: How were the phages purified? Was it precipitated? How stable were the phages here?

8. Line no. 67-68: What concentration? And 4% stain looks very high when compared to other studies.

9. Line no. 71-75: What concentration of phage? It should be explained at the start of all the experiments.

10. Line no. 154 and Figure 1: Need better TEM images.

11. Line no. 159 and Table 1: This phage is highly specific to one strain, especially since it did not infect the same bacteria from the same country and the host. How it is a good candidate phage for therapy, particularly in aquaculture applications. How big a cocktail can be?

12. Line no. 202: The numbers do not match the bacterial load and PFU. Please check.

13. Line no. 215: A siphovirus without tape measure protein is unnatural. Please cross-check the assembly and annotation.

14. The manuscript needs to be evaluated thoroughly for typos and English corrections. At this level, the manuscript cannot be published. 

Round 2

Reviewer 2 Report

I appreciate the author's involvement in making substantial revisions to the manuscript. There are some more concerns which need to be addressed before publication.

1. Line no. 101: What is the JEOL model? Is it 1200? Mention it.

2. Line no. 209: Write as siphovirus-like.

3. Figure 3: Burst size should be marked between 'the end of the latent period/start of the rise period and the start of the plateau period'. Correct the vertical arrow.

4. Genome analysis: The phage genome file is not released yet so I am unable to verify it. I have some basic questions. 1. Include the phage genome into the new family (see ICTV classification) https://link.springer.com/article/10.1007/s00705-022-05694-2, 2. Tailed phages without unidentified (CDS) tape measure protein is not acceptable or considered as a correct annotation because they are mostly conserved in the siphovirus-like phage group. 3. No CDS related to the lysis-related activity is annotated (endolysin-like, holin, spanin). Both the genome assembly and annotation need to be analysed. 4. Is the CDS gene call based on Shine-Dalgarno? Mention it. 5. How closely this phage genome is related to others in the database?

5. Line no. 392: Novel? How novel/new it is? There is no real comparison provided. To claim such a conclusion, at least provide a nucleotide similarity map using VIRIDIC (https://www.mdpi.com/1999-4915/12/11/1268). Is it a new phage within a new family/genus as per the recent ICTV release (see above)? 

Round 3

Reviewer 2 Report

The authors have made substantial revisions to the manuscript and it has been improved.

1. If possible, correct the annotation in the NCBI database. And in the published annotation file, vBVhaSMAG7_030 is annotated as tail fibre protein. Though the lineage is unclassification, based on the EM image it is classified as siphovirus-like so it better to annotated it.

Author Response

We have requested to update the annotation of the phage in NCBI and will be available very soon.

Specifically, we have requested for the following changes:

  1. vBVhaSMAG7_050 from coil containing protein to DNA circularization protein
  2. vBVhaSMAG7_055 from TMhelix containing protein to tail tape measure protein

vBVhaSMAG7_030 is indeed annotated as tail fiber protein in our annotation, we are not sure what would you like us to do with this annotation.